# Oral Health Strategies: Surveying the Present to Plan the Future of Community-Based Learning

**DOI:** 10.3390/healthcare11192646

**Published:** 2023-09-28

**Authors:** Nélio Jorge Veiga, Patrícia Couto, Patrícia Correia, Anna Carolina Volpi Mello-Moura, Pedro Campos Lopes, Maria José Correia

**Affiliations:** 1Universidade Católica Portuguesa, Faculty of Dental Medicine, 3504-505 Viseu, Portugal; pscouto@ucp.pt (P.C.); pcorreia@ucp.pt (P.C.); acmoura@ucp.pt (A.C.V.M.-M.); paflopes@ucp.pt (P.C.L.); mcorreia@ucp.pt (M.J.C.); 2Universidade Católica Portuguesa, Centre for Interdisciplinary Research in Health (CIIS), 3504-505 Viseu, Portugal

**Keywords:** oral health literacy, community-based learning, service learning, health promotion

## Abstract

**Introduction**: Oral health literacy relates to the capacity of individuals to acquire, understand and to act upon oral health information to make appropriate health decisions. This scientific review’s main goal is to analyze the strategies that improve oral health literacy within the community, specifically oriented to a community-based learning model focused on the most vulnerable risk groups in society. **Materials and Methods**: The current review is based on the literature on oral health literacy within community-based learning strategies. The present review selected scientific studies by searching MEDLINE and related databases, such as Web of Science and PUBMED, and by consulting existing bibliographies. **Results**: Based on the application of the inclusion criteria to the abstracts, 45 publications were retrieved which explicitly dealt with the definitions of oral health literacy, community-based learning, and service learning. Several studies have demonstrated that health consumers with low health literacy fail to understand the available health information. Therefore, innovative oral health literacy strategies should be undertaken. Service learning is an example of an educational approach where the student learns specific soft skills in the classroom and collaborates directly with an agency or institution and engages in reflection activities to deepen their understanding of what is being taught. **Conclusions**: One of the main strategies used to incorporate the oral health professional in social responsibility and direct contact in the community is through experiencing community-based learning projects. The future graduate must be an educator capable of educating patients in order to themselves acquire high-level skills in oral health literacy.

## 1. Introduction

Health literacy consists of the capacity of individuals to acquire, understand, and act upon health information to make the most appropriate health decisions for themselves. The main objective is the maintenance of health or the management of a disease or condition. Health literacy aims to empower individuals in the management of their health [1,2].

The term health literacy initially appeared in the 1970s and was considered to be a network that involves the individual, their families, and communities at the center of a training process that allows the improvement of prevention methods and the consolidation of healthy lifestyles [1,2,3]. 

The World Health Organization (WHO) defines health literacy as the acquisition of cognitive and social skills which permits an individual to gain the motivation and ability to understand information and knowledge related to the promotion and maintenance of good health [2,3]. 

Health literacy can be described on two main levels: personal and organizational. Personal health literacy corresponds to the level of capacity to obtain, process, and understand basic health information to make the most adequate health decisions. Organizational health literacy refers to the level of equity with which different organizations enable individuals to understand and use information for their own health-related decisions and actions [4]. 

Thus, the concept of health literacy is not limited to simply being able to read printed information and look for health services. The empowerment of the community is fundamental in improving daily habits and, consequently, health and quality of life, allowing individuals to acquire the capacity to participate in community health promotion and disease prevention actions [1,2,5,6]. Therefore, it is fundamental to assess the impact of functional health literacy to determine the capacity in which an individual must understand basic health information [7,8,9,10]. 

Low oral health literacy leads to a decrease in the adoption of adequate oral health daily habits. Good communication between patient and health professional will also permit an increase in the level of oral health literacy, leading to a decrease in the levels of anxiety during dental treatments and less reluctance to receive medical support [11,12]. 

For Horowitz and Kleinman, those with low oral health literacy, for example ethnic minorities, the elderly, and individuals with special needs, usually have a higher risk of disease [13,14,15]. The improvement in oral health literacy levels among the community can increase with the application of oral health promotion strategies [16]. 

Moreover, low health literacy is associated with an increased difficulty in reading or understanding certain content, which, simultaneously, reduces the application of this information in the improvement of health behaviors. It is fundamental to understand the association of low health literacy with higher risks of hospitalization, worse health status, greater adherence to therapy, and higher health expenses.

Knowing the association between the levels of autonomy in health literature and health outcomes, individuals should play an active role in the management of their own health. As a result, efforts have been made at the local, national, and global levels to promote population health literacy [15,16].

This scientific review’s main goal is to analyze the strategies that improve oral health literacy within the community, specifically oriented to a community-based learning model focused on the most vulnerable risk groups in society.

## 2. Materials and Methods

The current review is based on the literature on oral health literacy within community-based learning strategies. The present review selected scientific studies by searching MEDLINE and related databases, such as Web of Science and PUBMED, by bibliographic consultation. The forward/backward reference chaining technique was used, and recent activities in health literacy were tracked. To retrieve studies, 12 keywords (health literacy, oral health literacy, community oral health, community-based learning, service-learning, public health, communication, information, strategies, health promotion, education, healthcare) were combined using the Boolean operator AND with the search terms “health literacy”, “community-based learning”, and “service-learning”. Studies were selected for inclusion in the review based on their abstracts. Eligible studies were included considering the following inclusion criteria: (1) written in English; (2) concerned with oral health literacy; and (3) offering relevant content regarding the definition of health literacy and community-based learning, or a combination of both.

## 3. Results

The combination of the keywords resulted in the initial identification of a total of 233 publications. An additional number of publications was found using the reference tracking method and, therefore, were included in this review. Based on the application of the inclusion criteria to the abstracts, 45 publications were included, permitting the development of the present review. The types of studies analyzed were mainly epidemiological observational studies, followed by case and experimental studies, narratives, and systematic reviews and policy papers developed towards the improvement of oral health promotion strategies (Table 1).

## 4. Discussion

### 4.1. Defining Oral Health Literacy Strategies in the Community

Oral health literacy is a relatively recent issue in the medical and dental literature. Oral health literacy has a considerably high impact on health outcomes, and specific strategies and actions will lead to the improvement of health levels among the population. 

A person should develop the ability and self-consciousness to obtain information related to healthcare. Several studies have demonstrated that health consumers with low health literacy fail to understand the available health information, which would increase health levels and, consequently, their quality of life. Therefore, one of the major health strategies must be focused on the positive interactions between modern healthcare and the population, to establish an integrated health promotion plan focused on disease prevention [18]. 

This can be obtained through the development of oral health educational strategies involving all fields among the community, from schools and social support institutions to dental clinics and health centers [18].

Improvements have been made in the field of oral health in past years; however, the journey is still at the beginning and a lot of work has yet to be accomplished. This can be achieved by understanding the social and economic barriers placed on care providers and patients. Achieving higher improvement levels in oral health literacy will always require the motivational effort of oral health professionals and should be focused on an individual patient-centered educational and promotional approach [1]. 

There have been defined specific oral health literacy dimensions, according to some studies, such as basic and specific oral health care system knowledge; marketing and consumer behavior; and oral health competencies at the workplace and political engagement [36,37]. 

Health literacy will increase the interest in receiving health information, which will permit a higher interest in accepting health information and help an individual to participate in shared decision-making towards healthcare issues [1,2,37].

There are various types of scales used to measure the impact of oral health literacy and its association with oral health status and quality of life. The Rapid Estimation of Adult Literacy in Dentistry (REALD 30) is an example of how oral health literacy can be measured. The application of this scale is relevant to the point that it has been validated in several countries worldwide [17]. 

Knowledge about the determinants of health is deeply important to justify the implementation of developed oral health strategies directed to the community [30]. This is an important way for a health professional to understand the direction in which oral health promotion should be conducted by the individual and at the community level [30]. Research on the perception of the oral health status of a community is fundamental, specifically regarding the most vulnerable. An example is the assessment that can be developed among the elderly by the application of the Geriatric Oral Health Assessment Index (GOHAI) [38]. 

There are numerous factors which can have a considerable impact on low health literacy levels among the community, such as a lower use of primary preventive services and treatments and an inability to understand prescriptions and self-care instructions [23]. For instance, among children, parents’ knowledge on oral health literacy is the key predictor in reducing social inequalities in health, through actions undertaken at a community scale. The prevention of early childhood caries needs a combination of generic and targeted interventions [23]. 

Nonetheless, we must consider the application of the latest technologies in oral health literacy. The development of applications for computers, tablets and cell phones has been growing exponentially, as it allows a more efficient learning environment and can reach all ages and territories in the future. This will certainly be a path of scientific progress towards digital solutions both on conceptual and practical levels, with the goal of empowering the community towards healthy lifestyles and, consequently, leading to an increase in quality of life [23,39].

### 4.2. Which Guidelines Are More Appropriate for Healthcare Literacy?

In 2016, the Survey on Health Literacy in Portugal (ILS-PT 2014) was published, which revealed that Portugal has a low prevalence of citizens with an excellent level of health literacy (8.6%) when compared to the average of eight other European countries (16.5%). This study also showed that almost half of the population (46.3%) has insufficient levels of health [32]. 

People with poor health levels may present serious difficulties in understanding simple instructions, which makes it difficult to apply the health information received. Difficulties are related not only to the language complexity level, but also extend to the understanding of the words, images, graphics, and schemes used in healthcare documents [31,32]. 

The elderly are a particularly vulnerable group, presenting low levels of health literacy. It is important to emphasize that low health literacy crosses all socioeconomic groups and academic levels [15,31]. 

Direct contact with health professionals is the most highly recommended way of providing health information. Therefore, it is essential to increase care and communication strategies, particularly with people with low health literacy. Communicating in the healthcare setting, whether orally or through written materials, can be adapted to the population [40]. 

Communicating is the simple act of transmitting, and not only distributing information. It implies an exchange between the sender and the receiver of this information. It also implies knowing who the person receiving the information is, in what context they receive it, and what will be the result of the interaction. It is up to the health professional to listen and adjust the content to the person and their level of literacy, and assess what they have understood. Understanding is a fundamental step that allows adjustment in the moment and the assessment of a true communication process. The “teach-back” is an example of true communication at the literacy level. It consists of explaining when using certain information and then asking the person if he/she understood using the person’s own words. This gives the health professional the opportunity to clarify some points and gives the patient the opportunity to consolidate knowledge [34]. 

The main tools that can be applied and developed to be a part of oral health literacy strategies are:▪Modern technologies that can help in service-learning processes;▪In-person and on-line meetings and courses for health promotion and empowerment;▪E-books and open access journals for health information;▪Specific apps for oral health literacy teaching aimed towards caregivers, the elderly and individuals with special needs [21].

These tools should be applied towards the main risk groups that can benefit more with the definition of specific communication tools to improve their empowerment towards oral health literacy. These are mainly:▪Low-income individuals and families;▪Individuals with special needs;▪Institutionalized and non-institutionalized elderly people and their respective caregivers;▪Migrants [21].

Research in oral health has recently developed efforts to study and understand the best strategies to communicate and improve oral health outcomes among communities that present lower levels of oral health literacy [34,40]. A key goal of improved communication to increase oral health literacy is the use of simple language. Simple and plain language should present clear communication and offers to healthcare professionals an opportunity to focus on a patient-centered communication strategy [35,41]. Oral health professionals and all team members should be well prepared to create a patient-centered communication environment focused on patients that present literacy and language needs [35,41]. 

Communication between health professionals and users should serve as a basis for decision-making, focused on therapeutic, preventable, and behavioral levels. To maximize communication within society, it is essential to adopt strategies that improve the understanding of vulnerable people with less access to health literature [25].

### 4.3. Service-Learning: An Example of a Community-Based Project in Dental Medicine

One of the main examples of a community-based project applicable to dental students is Service learning. Service learning is an educational approach, student-centered, that allows the participants to learn specific soft skills in the classroom and, simultaneously, collaborate directly with an agency or institution (usually a non-profit or social service group) and engage in specific reflection activities to deepen their understanding of what is being taught [19]. 

Service learning is a form of teaching that permits the combination of educational methodologies and community service experiences. It represents a holistic approach that permits the establishment of an important link between students and the institutions that serve the community, and more specifically the most vulnerable communities [42]. 

A community-based learning course provides the opportunity for dental students to develop valuable contributions towards specific communities through the active participation in organized service experiences which are coordinated with the school and community/institutions; specific moments for the students to think, talk, or write about their experiences; the acquisition of academic skills and knowledge in real life situations; and the development of a sense of caring for others and social responsibility [27,33,43]. 

In dental medicine, and more specifically in community oral health, the development of service learning, based on community-based learning, is focused on oral health literacy among the most vulnerable of the community [24,27,33]. 

From the elderly and their caregivers to patients with special needs, service learning can develop work-readiness skills towards social responsibility during the course through increased contact with the basic problems of the community in different environmental contexts [31]. Students can develop community-based service experiences to complete the objectives of a service learning project. Structured reflection is also developed in these projects, mainly focused on oral health literacy and disease prevention. The university–community partnership is essential to establish ongoing relationships between the university and community partners in which students are involved in service [15,28,44]. 

Regarding students, the literature is in agreement on the positive impact of service learning in numerous fields, namely academic performance, social commitment, reflective and communication skills, professional development, team spirit, and the increase in values and ethical principles of respect, collaboration, and coexistence with others [24,29,45].

Thus, we can verify that the service learning methodology promotes learning and service in the community context through the application of skills and the reflective understanding of the curricular content integrated in the respective curricular units, having as a main objective the development of a sense of social responsibility, inclusion, and social benefit for the future [26]. 

It is essential to analyze the perceptions of those who receive the service, mainly the caregivers and directors of the partnership institutions, as well as the perceptions of those who guide this pedagogical methodology, such as the teachers of the various curricular units that are part of the service learning project. The goal is that dental students have the possibility of becoming clinicians who develop their profession among the community and identify oral health disparities related with actual social determinants of health, becoming entirely focused on their social responsibility as health professionals [20,22,24]. 

## 5. Conclusions

One of the main strategies to incorporate the oral health professional in social responsibility and direct contact in the community is through them experiencing community-based learning projects during their training phase. The future graduate must be an educator capable of educating patients to themselves acquire high-level skills in oral health literacy. It is important to highlight that oral health-related degrees’ curricula should favor community-based learning as a successful and engaging learning model that addresses both professionalism and humanism.

## Figures and Tables

**Table 1 healthcare-11-02646-t001:** Studies used and characterization of each according to the main conclusions mentioned in each one.

Author	Date	Type of Study	Health Literacy Level	Main Conclusions
Costa H et al. [17]	2022	Cross-sectional study	Personal health literacy	The new scale can be applied to assess oral health literacy among older Portuguese adults.
Berezovsky B et al. [18]	2021	Narrative review	Organizational health literacy	Implementing preventative measures and developing potentially better ones should be intensified if the disease burden of oral diseases worldwide is to decline in the future.
Lin TH [19]	2021	Case study	Personal health literacy	Various reflection activities triggered different levels and functions of reflection by the participants.
Nilsson A et al. [20]	2021	Scoping review	Organizational health literacy	This review has highlighted the need for national and international guidelines to ensure the mandatory inclusion of sufficient and specific gerodontology training to prepare graduates for a growing dentate-frail and care-dependent population.
Khurshid Z et al. [21]	2021	Review	Organizational health literacy	All the essential information related to the generally used electronic databases in dentistry research. This will be helpful for dental students, residents, consultants, and allied science researchers.
Goodman XY et al. [22]	2020	Review	Organizational health literacy	Share of strategies used to design the instruction sessions, reflections on teaching these themes, lessons learned, and suggestions for other liaison librarians who might have an interest in teaching about cultural competence or cultural humility.
Marquillier T et al. [23]	2020	Scoping review	Personal health literacy	Parents’ knowledge and oral health literacy are the key predictors to be preferentially targeted in view of reducing social inequalities in health through actions undertaken on a local scale.
Claiborne DM et al. [24]	2020	Cross-sectional	Organizational health literacy	The service learning educational project has potential for easy integration within dental hygiene and advanced practice nursing curricula.
Yimenu DK et al. [25]	2020	Cross-sectional	Organizational health literacy	Health professionals should master basic oral health-related knowledge, and they should practice basic oral health care practices to become role models for their patients.
Ruiz-Montero et al. [26]	2019	Systematic review	Organizational health literacy	Service learning can have a positive effect on older adults’ health promotion and can enhance their community participation.
Nierenberg et al. [27]	2018	Cross-sectional	Personal health literacy	These findings suggest that exposure to patients who lack dental care and have severe oral health problems can impact developing nursing and dental professionals in ways that can increase their appreciation of interprofessional practice and their future willingness to provide care in underserved settings.
Phlypo I et al. [28]	2018	Experimental study	Personal health literacy	Findings demonstrated that within a semester, it was possible for students to obtain a limited but positive impact on the local community.
Blaizot A et al. [29]	2018	Cross-sectional	Organizational health literacy	The study provided a foundation for building future orientations in health care for patients with limited decision-making abilities.
Ghaffari M et al. [30]	2018	Meta-analysis	Organizational health literacy	Oral health education and promotion interventions have effective and positive impacts on dental visits, attitudes, and brushing and flossing behaviors for 3 months post-intervention among children.
Baskaradoss JK et al. [9]	2018	Cross-sectional study	Personal health literacy	Improving the OHL of patients may help in efforts to improve the adherence to medical instructions, self-management skills, and the overall treatment outcomes.
Firmino R et al. [12]	2018	Systematic review/meta-analysis	Personal health literacy	No association exists between OHL and any of the outcomes investigated in oral health behaviors, perception, knowledge, and dental-treatment-related outcomes.
Nihtila A et al. [31]	2017	Experimental study	Organizational health literacy	Multiple approaches based on individual needs are required to improve the oral health of vulnerable older adults, including integrating dental preventive care into the daily care plan carried out by home care nurses.
Espanha R et al. [32]	2016	Cross-sectional study	Personal health literacy	The literacy level in Portugal is very similar to other countries in the European survey and information sources are an important instrument to improve health literacy.
Coe JM et al. [33]	2015	Cross-sectional	Organizational health literacy	It can be concluded that community-based service-learning positively impacts the attitude of last year’s dental students toward understanding needs at the community level and the attitudes towards helping to providing dental care at the community level.
Haun JN et al. [6]	2015	Retrospective study	Personal health literacy	Lower health literacy is a significant independent factor associated with increased health care utilization and costs.
Holtzman JS et al. [16]	2014	Retrospective study	Personal health literacy	Individuals who use fewer sources of oral health information, a subset of health literacy skills, are more likely to fail to show for dental appointments.
Guo Y et al. [11]	2014	Cross-sectional study	Organizational health literacy	Improved patient–dentist communication is needed as an initial step in improving the population’s oral health.
Hongal S et al. [34]	2013	Narrative review	Personal health literacy	Poor health literacy is considered to be a contributor to poor oral health status in an individual, poor health outcome in a community, and health inequalities.
Sorensen K et al. [2]	2012	Systematic review	Organizational health literacy	A model is proposed integrating medical and public health views of health literacy.
Shokrya AE et al. [15]	2011	Quasi-experimental study	Organizational health literacy	The oral health educational program is effective in improving the elderly’s quality of life.
Holtzman JS et al. [35]	2009	Cross-sectional	Personal health literacy	These changes in attitude may reflect the students’ greater understanding of the complexity of the determinants of oral health because of their community education experiences.
Horowitz A et al. [13]	2008	Narrative review	Personal health literacy	Effective communication with patients is the cornerstone of quality dental care and oral health outcomes.
Lee JY et al. [8]	2007	Cross-sectional study	Personal health literacy	Dental health literacy may be distinct from medical health literacy and may have an independent effect on dental health outcomes.
Nutbeam D. [1]	2000	Review	Organizational health literacy	Improving health literacy in a population involves more than the transmission of health information, although that remains a fundamental task.

## Data Availability

The data and additional information used to generate and support the review established are available from the corresponding author upon request.

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
