# Peer review of "Oral Health Strategies: Surveying the Present to Plan the Future of Community-Based Learning"

_healthcare, 2023, doi:10.3390/healthcare11192646_

Round 1
Reviewer 1 Report
This this a review of the manuscript asa opinion entitled : Oral health strategies: surveying the present to plan the 2 future of community-based learning
The Authors made a review work and the paper was prepared as an opinion.
The most importatnt conclusion showed a need of incorporation the health professionals in social responsibility and direct contact in community by experiencing Service-Learning programs during their training phase.
This present conclusion kicks forward a voice in discussion how to improve oral health in communities. Based on the selected literature the Authors concluded that there are changes needed to improve the quality of oral health level. There are many risk groups that have to be covered in every community such as elderly, their caregivers, patients with special needs, also patients effected from chronic and acute general diseases.The goal is starting from the practical phase during University education of dental health professionals. In my opinion the article is going in right direction but it would be great to show any graphical presentation of the most significant directions, for example the Authors underlined communication as important tool in working with risk groups, in addition the Authors could prepare a list of risk groups that have to be included to abovementioned program? Next it is worthnoting any modern technologies that can help service-learning, like in person or on - line meetings, e-books, open access journals for health information?
Did the Authors meet any limitations of their study?
After fullfilling the most desired open questions the Opinion prepared by the Authors may be accepted for further consideration by Editorial Board.
Author Response
Thank you for your suggestions. We have made the changes directly in the manuscript. We did not find significant limitations to the research, because we were able to find the appropriate scientific articles to develop the text.
Reviewer 2 Report
Thank you for this article, which covers the important concept of health literacy, specifically oral health literacy. Although the article's introduction is well written and succinctly explains the background to the concept, the remainder of the article is not well structured. Specifically, the results section is hugely lacking. There is no explanation as to what type of studies were examined, where the studies were carried out, what populations were examined, what type of interventions were carried out (if applicable). Based on this, there is not enough substance in the results section for this article to add significantly to the literature. I understand that this is an opinion piece but I believe further information needs to be given regarding the studies carried out.
The English is good overall, however there are some areas that would benefit from re-examination. E.g. Line 45- "complementary" is not the correct word here. Line 62/63 are not very well phrased.
Author Response
Thank you for the suggestions. All the changes have been made in the manuscript. We have added to the results a Table which refers to the main studies analyzed and that where used in the development of the present manuscript.
Reviewer 3 Report
The authors of the article have introduced the problem of what is agreed to define literacy in oral health. The literature review provides eligible explanations insofar as it is limited to oral health. However, the content of the article is confusing because it distinguishes between health and oral health. Therefore, the authors adopt an approach obviously oral health practitioners centred preferably to an approach on patients centred, who are the central partners in the education process in oral health literacy. This leads to structuring the curriculum by introducing the idea that among the skills acquired by future oral health practitioners, it should be added that the future graduate must be an educator capable of educate the patients in order to themselves acquire high level skills in oral health literacy.
Service-Learning: a community-based project in dental medicine (line 193) is solely students centred. Patients are totally absent from the process.
The authors are invited to consult these links
https://pubmed.ncbi.nlm.nih.gov/?term=%28Mayeaux%5BAuthor%5D%29+AND+%28Murphy%5BAuthor%5D%29&sort=
https://pubmed.ncbi.nlm.nih.gov/16100861/
And the attached file
Regards

Author Response
Thank you for the suggestions made. We have made the indicated corrections in the manuscript in order to improve it for future publication.
Reviewer 4 Report
Dear authors,
Thank you for your effort. In order to provide you with some helpful comments for improving the quality of the paper, I would suggest you proceed to the following:
1. Please use instruction for authors for references
2. In the section of methodology, you should incorporate the relevant prisma chart flow of your review articles
3. You also need to form a table with all the studies you used (names of authors, year of publication, methodoloy, main conclusions from each article etc.) and you should characterize each one of them according to the discussion issues you are mentioning further.
4. "Results " should be changed according to the table's main findings. in that case you delete discussion title of your findings.
5. Findings are a bit chaotic. I could not follow them. You should rearrange them according to table.
5. Discussion part should have an overview of what you found from the review and suggestions about policy makers on the issue. You should not repeat or rewrite what is in the previous sections. you should also write about limitations of your approach (this is not a systematic review) and you shouls suggest further research projects on this interesting issue.
6. Rearrange accordingly the conclusions
7. Mention on your title that this is a narrative review and not a systematic one.
Good luck!
The reviewer
Should be checked!
Author Response
Thank you for the suggestions made. The proposed manuscript is not a systematic review. It is defined as a narrative review.
We have added a table with the characterization of studies used for the development of the manuscript and have made the proper corrections in the manuscript.
Round 2
Reviewer 2 Report
Table 1: Costa study – What is the “new scale”?
I would like to draw your attention to the conclusion you found in the Tabrizi study “Dental education in many U.S. dental schools may provide adequate education and create competent general dentists.”- This is not relevant to your study. You are looking at oral health literacy, this is about training general dentists, presumably they mean clinical training, unless otherwise stated.
I feel that the discussion section is still too convoluted and not directly relatable back to the results. You say that your “main goal is to analyze the strategies that improve oral health literacy within the community, specifically oriented to a community-based learning model, focused on the most vulnerable risk groups of society”. However, the only strategy you discuss in depth is Service Learning. Therefore, a more legible but in-depth article would possibly have been to just examine studies that discussed Service Learning. I feel that you have “cast your net too wide” in the search and have not properly discussed all the results that you found.
Some small changes to be made, e.g. line 201; none-institutionalized should be "non-institutionalized"
Author Response
Thank you for your review. We have accepted the suggestions.
I have corrected in Table 1 the study of Costa et al., and changed the conclusion to "The REALD30 scale was validated in order to be applied to assess oral health literacy among older Portuguese adults."
We eliminated of Table 1 the Tabrizi study “Dental education in many U.S. dental schools may provide adequate education and create competent general dentists.” We entirely agree with the suggestion.
Our main objective was to explain the importance of community-based learning methods for future oral health professionals. We indeed focus more Service-Learning as an example of a method that is being widely applied in health courses. In the manuscript, we have changed the section of Service-Learning, referring more as an example.
The change on line 201 was made and we revised the English.
Reviewer 3 Report
The manuscript has been sufficiently improved to warrant publication in Healthcare.
Author Response
Thank you very much for all the suggestions made in order tom improve this article.
Reviewer 4 Report
Dear authors
Thank you for this interesting opinion.
Table with studies is an excellent way for the reader to receive information. Try to put more information on their conclusions as some articles seem irrelevant. a good idea would be to put another column reporting on which one is focus on patients (personal) and which to organizational literacy.
In the discussion part you should try not to repeat the conclusions of the articles you used. You rather make your own suggestions that can be reinforced by conclusions of others
In this sense conclusions should be rewritten and expanded to more than "Service Learning"..
Grammatical mistakes are here and there in the text. Please see carefully .
Good luck
minor grammatical mistakes, minor corrections needed
Author Response
Thank you for the suggestions.
We have considered all of the them and in the manuscript and we have added a new column in the table with the focus of the study: personal or organizational health literacy.
We have also rewritten the Conclusions in order not to seem like the Discussion and the Conclusions are not, at the moment, limited to Service-Learning.